# Learning to kill: Why a small handful of counties generates the bulk of US death sentences

**Frank R. Baumgartner**[1]*, **Janet M. Box-Steffensmeier**[2], **Benjamin W. Campbell**[2], **Christian Caron**[1], **Hailey Sherman**[1]

**1** Department of Political Science, University of North Carolina at Chapel Hill, Chapel Hill, North Carolina, United States of America, **2** Department of Political Science, Ohio State University, Columbus, Ohio, United States of America

* frankb@unc.edu

**Data Availability Statement:** All relevant data are within the manuscript and its Supporting Information files.

## Abstract

We demonstrate strong self-referential effects in county-level data concerning use of the death penalty. We first show event-dependency using a repeated-event model. Higher numbers of previous events reduce the expected time delay before the next event. Second, we use a cross-sectional time-series approach to model the number of death sentences imposed in a given county in a given year. This model shows that the cumulative number of death sentences previously imposed in the same county is a strong predictor of the number imposed in a given year. Results raise troubling substantive implications: The number of death sentences in a given county in a given year is better predicted by that county's previous experience in imposing death than by the number of homicides. This explains the previously observed fact that a large share of death sentences come from a small number of counties and documents the self-referential aspects of use the death penalty. A death sentencing system based on racial dynamics and then amplified by self-referential dynamics is inconsistent with equal protection of the law, but this describes the United States system well.

## Introduction

Imagine a death penalty that is imposed on killers in different areas of the country, or even across localities within individual states, in a manner that is substantially random but, to the extent that any systematic patterns are apparent, those are related to ugly racial dynamics, including the legacy of racial violence in decades past. Then consider the possibility that each locality settles into a pattern of use or avoidance of the death penalty based on its own accumulated history. In such a self-reinforcing system, small initial differences across counties would eventually accumulate into vast differences, with some localities virtually never using the death penalty and others using it much more frequently. Mostly, these differences would be unrelated to such factors as homicide rates, but to the extent that any statistical patterns could be discerned, two things would stand out: racial dynamics, since these were part of the dynamic

**Funding:** The authors received no specific funding for this work.

**Competing interests:** The authors have declared that no competing interests exist.

that set the localities onto their different paths in history, and vast differences in use. The self-reinforcing dynamic would generate a "stretched" distribution of use with a few outlier counties using the punishment much more than others, and the vast majority not using it at all. The racial dynamics would still be apparent in the final distribution, however: The high-use counties would disproportionately come from those counties with histories of racial violence against African-Americans. Of course, counties with more people and more homicides might have more death sentences, but this linkage would be attenuated by racial and self-reinforcing dynamics.

The United States (US) Supreme Court ruled the death penalty system unconstitutional in 1972 because of concerns about patterns similar to these. Justice Potter Stewart wrote that the small number of individuals chosen for the penalty of death represented a "capriciously selected random handful" and that such a system is "cruel and unusual in the same way that being stuck by lightning is cruel and unusual." Further, he noted that "if any basis can be discerned for the selection of these few to be sentenced to die, it is the constitutionally impermissible basis of race." However, he considered the issue of racial bias not to have been proved, so he "put it to one side" (see *Furman v. Georgia* (408 U.S. 238 (1972), 309–310). In invalidating the US system of capital punishment in 1972, the justices were therefore concerned about two things: racial bias, which they suspected, and capriciousness, which they found. We focus on the question of geographical concentration of the death penalty in just a handful of localities, and we seek to answer the question: Why these counties but not others? The answers reveal racial dynamics combined with caprice, exactly what the Justices were concerned about in 1972.

We base our analysis on every death sentence imposed in the US from 1972 through the end of 2019 and present two distinct empirical tests. These show that, if there is any statistical pattern, it is indeed race. Further, they show that the system is dominated by a self-reinforcing system that, over more than 45 years, has generated a capricious and arbitrary distribution where the number of homicides is only loosely related to the number of death sentences. The better predictor of whether a given county will sentence an individual to death in a given year is not the number of homicides in that county in the previous year, but rather the number of death sentences that county has previously imposed. A district attorney's office may or may not accumulate the skills, knowledge, and practice needed successfully to carry out a capital trial leading to a death sentence. Whether or not previous decades of experience have led to these skills and practices, however, is unrelated to the heinousness of the next crime that may occur within any given county. Therefore, it should be unrelated to the odds of seeking or imposing a death sentence. But in fact, it is one of the most powerful and consistent predictors.

## A puzzle: The geographic distribution of death sentences

As of 1972, 41 states, the District of Columbia, the federal government, and the military had a legal death penalty, for a total of 44 jurisdictions. Following *Furman*, which ruled all of these systems inoperable, most of these states quickly reestablished their capital punishment systems with further safeguards to ensure "proportionality" so that the "modern" death penalty would avoid the flaws, particularly capriciousness, that the Justices had noted in *Furman*. By the end of 1976, 35 states had reestablished; the number rose to 40 by 1984 and stayed roughly at that level until a series of abolitions beginning in 2007. Since then, New York, New Jersey, New Mexico, Illinois, Connecticut, Maryland, Delaware, Washington, New Hampshire, and Colorado have ended their death penalty systems, bringing the number of retentionist jurisdictions as of 2019 to 29 (see [1], pp. 11–12; [2]). In the analyses below, we include only states allowing the death penalty in the year of analysis. Because we focus on the geographical variability in

the use of the death penalty, we exclude the US military (which has sentenced 15 individuals to death since reestablishment in 1984, but carried out no executions) and the federal government (which has issued 79 death sentences since reinstatement in 1988, and carried out three executions). (These numbers are current as of the close of 2019.)

We are concerned here with the geographical concentration of the death penalty in just a few jurisdictions. This has previously been noted by many scholars and activists, so we take it as a starting point (for fuller discussions of this concentration, see [3–6]). Because our focus is on counties (within states), we must first note that many counties are small with regards to population, but a few are very large. Any discussion of geographic concentration of the death penalty must start with this baseline. Obviously, there would be no surprise if Los Angeles County, California (2010 population: 9,840,024) had more homicides or death sentences than Loving County, Texas (2010 population: 85).

Table 1 shows the top 25 counties in the nation with regards to cumulative death sentences. Two counties stand out: Los Angeles, California and Harris, Texas, with 311 and 299 death sentences, respectively, more than the vast majority of states. The table also lists the rate of death sentences per 100 homicides and the rate of homicides per 100,000 population. If there was a direct link among these variables, we would expect some consistency here. But we see

**Table 1. Top 25 death sentencing counties, with cumulative homicides.**

| County | Homicides | | Death Sentences | | Death Sentences per 100 Homicides | | Homicides per 100,000 Population | |
|---|---|---|---|---|---|---|---|---|
| | Number | Rank | Number | Rank | Rate | Rank | Rate | Rank |
| Los Angeles CA | 56,112 | 2 | 311 | 1 | 0.55 | 633 | 13.76 | 87 |
| Harris TX | 23,465 | 5 | 299 | 2 | 1.27 | 395 | 16.03 | 51 |
| Philadelphia PA | 17,851 | 6 | 187 | 3 | 1.05 | 438 | 24.10 | 13 |
| Maricopa AZ | 10,746 | 11 | 179 | 4 | 1.67 | 312 | 8.90 | 300 |
| Cook IL | 36,984 | 3 | 157 | 5 | 0.42 | 670 | 15.31 | 59 |
| Miami-Dade FL | 14,035 | 8 | 118 | 6 | 0.84 | 534 | 14.75 | 68 |
| Clark NV | 5,874 | 21 | 118 | 7 | 2.01 | 236 | 11.18 | 163 |
| Oklahoma OK | 3,932 | 39 | 116 | 8 | 2.95 | 123 | 13.34 | 96 |
| Riverside CA | 4,636 | 30 | 110 | 9 | 2.37 | 190 | 7.22 | 413 |
| Duval FL | 5,138 | 27 | 110 | 10 | 2.14 | 220 | 15.47 | 57 |
| Dallas TX | 14,391 | 7 | 107 | 11 | 0.74 | 577 | 15.70 | 54 |
| Cuyahoga OH | 7,797 | 15 | 90 | 12 | 1.15 | 418 | 12.02 | 133 |
| Orange CA | 4,417 | 31 | 82 | 13 | 1.86 | 269 | 3.83 | 744 |
| Broward FL | 4,645 | 28 | 80 | 14 | 1.72 | 296 | 7.27 | 410 |
| Hamilton OH | 3,315 | 47 | 80 | 15 | 2.41 | 183 | 8.48 | 324 |
| Jefferson AL | 5,330 | 25 | 78 | 16 | 1.46 | 356 | 17.57 | 42 |
| Bexar TX | 7,690 | 16 | 77 | 17 | 1.00 | 452 | 12.55 | 120 |
| Hillsborough FL | 3,861 | 40 | 75 | 18 | 1.94 | 251 | 9.02 | 287 |
| Tarrant TX | 5,840 | 22 | 74 | 19 | 1.27 | 396 | 9.55 | 258 |
| Shelby TN | 8,064 | 14 | 74 | 20 | 0.92 | 500 | 20.56 | 19 |
| Pima AZ | 3,123 | 51 | 68 | 21 | 2.18 | 215 | 9.14 | 280 |
| Pinellas FL | 2,188 | 69 | 64 | 22 | 2.93 | 125 | 5.69 | 550 |
| Alameda CA | 6,308 | 20 | 63 | 23 | 1.00 | 454 | 10.24 | 220 |
| San Bernardino CA | 6,719 | 17 | 62 | 24 | 0.92 | 491 | 9.76 | 241 |
| Orange FL | 2,971 | 56 | 54 | 25 | 1.82 | 273 | 8.03 | 353 |

Note: Only counties with more than 100 cumulative homicides over the relevant time period are included.

very little. In fact, the correlations are surprisingly low; in fact, the rate of death sentences per 100 homicides and the rate of homicides per 100,000 population correlate at -0.12. The counties with the highest raw numbers of death sentences listed in the table include not a single county that ranks in the top 100 with regards to death sentences per 100 homicides. And, while Philadelphia ranks 13th in terms of homicides per 100,000 population, the counties listed here are not, in general, the most homicide-prone in the nation, as the table makes clear. Many high-homicides counties are absent from the list of top death sentencing counties, despite the fact that homicides are included only for those years where the death penalty was a legally available option. In sum, Table 1 presents a puzzle. What is going on?

What process would make the number of death sentences so concentrated in just a few jurisdictions? Los Angeles and Houston (Harris County) are high on both the lists of homicides and death sentences, but consider Atlanta (Fulton County), Georgia. It is high on the list of homicides, but has only a total of 16 death sentences. Phoenix (Maricopa County), Arizona, had slightly more homicides than Atlanta, but 179 death sentences. Baltimore had about the same number of homicides (roughly 10,000 over the period), but just six death sentences. There is little reason to think that homicides would be more heinous or deserving of death if they occur in one place rather than another. But when we look at different places with roughly similar numbers of homicides, we see widely divergent paths with regard to the use of the death penalty. This pattern is the same when we look across counties within a given state, so cannot be attributed solely to differences in what crimes are death eligible, which varies across states (Many states have broad death eligibility laws, including such things as any homicide occurring during the commission of an underlying felony (such as a robbery). Others, such as New York during the time it had the death penalty, had more narrowly targeted eligibility rules.). We turn in the next section to present our answer to this puzzle: self-reinforcement. Following that, we present two distinct empirical tests.

## Imposing a death sentence

When a homicide occurs, police investigate, and the district attorney brings charges. In states with a valid capital punishment statute, procedures vary but all have in common that the state must decide whether to "seek death." Typically, the state may seek death only if the crime meets certain statutory criteria (e.g., it is a death-eligible crime as defined in the statute). But within the category of death-eligible crimes, the district attorney has discretion to seek death or not. This is the first, and generally most important, step in the process. (North Carolina's law required district attorneys to seek death in all cases where the crime was capital eligible until 2001, when discretion was granted. Since 2001, every death state has afforded district attorneys such discretion.) Capital trials have two stages: guilt and punishment. The same jury sits for both, and if the defendant is found guilty of a capital crime in the first stage, then the jury sits for the "penalty phase" to consider aggravating and mitigating evidence, and pronounce a decision. In most states, the jury's decision is binding, but in some states, the judge decides or may overrule the jury.

## A theory of self-reinforcement

As some of us have previously described, the death penalty process is local and self-referential (see [7, 8]). Legal scholar Lee Kovarsky [5] describes it as the development of local "muscle memory": Localities either get good at the complex process of bring a capital case to its conclusion, or they do not. Brandon Garrett and colleagues [6] describe a process as follows:

> Once an office assembles a staff that has handled a capital trial, it draws upon this capacity to pursue the death penalty in subsequent cases, which further augments the office's institutional capacity to pursue the death penalty. This self-reinforcing dynamic between capacity and caseload makes it more likely for offices that obtain death sentences to seek the death penalty going forward. Conversely, offices that cease to obtain death penalties (or never obtained death penalties in the first place) may be less likely to reverse course as institutional capacity for death penalty sentencing erodes (or is never developed). . . . This path dependency may reflect practices of prosecutors who make the charging decisions whether to seek the death penalty, but it may also capture defense lawyering, judges, jurors, and other features of a county that make it more likely to continue to death sentence over time
>
> (p. 600).

Garrett and colleagues demonstrate strong support for these ideas with a large analysis similar to the second test we present here, but only for the period of 1990 through 2016. We build on their important work, extending it in time to cover the full post-Furman period, basing it on an independent data collection effort, revising the statistical test to account for the over-abundance of zeros in the data, and adding to it our first statistical test, one of event dependency.

Criminal justice is clearly a state function, so procedures differ from state to state. Many features differ by state: what crimes are capital-eligible, for example. In those states with the death penalty, when a capital crime occurs, the district attorney typically chooses whether or not to seek death. We control for state in all models below and focus on county-level variability within state. There is, of course, no reason to expect death sentencing rates to be identical across counties. First, within the same state, some counties may randomly see slightly higher or lower numbers of capital-eligible homicides as a share of all homicides. Second, there would likely be stochastic variability in the odds that the police investigation isolates a suspect and provides enough evidence to make a case "beyond a reasonable doubt" in court. Similarly, services available to indigent defendants might differ from place to place, as would the ideology of judges, district attorneys, and jurors. (Indeed, in Table 3 below we incorporate a measure of citizen ideology, and this shows almost a perfect normal distribution as it varies across states and time. Such stochastic variation cannot directly explain the high concentration that we observe in death sentences across localities.) There is no reason, therefore, to expect uniformity in death sentences as a share of all homicides. Stochastic variability would naturally generate some random differences in these rates.

Certain factors associated with capital prosecutions, on the other hand, can be self-referential, not randomly distributed. Consider the question from the perspective of the first mover, the district attorney. Given a new capital eligible crime, should the district attorney's office "go for death"? One relevant concern would be fairness. Was this crime as bad or worse than any previous crime for which the same office previously sought capital punishment? If not, then the death penalty for this crime might be considered inappropriate because it is excessive compared to previous cases. If, on the other hand, the crime was worse than others where the death penalty had been sought, then a capital prosecution might seem to be required on the basis of historical consistency and fairness. Another consideration is the odds of winning: will the jury vote for death, and will the judge agree? If not, then the costs, time commitment, and effort spent seeking a death penalty might be misplaced; in most states the district attorney can seek a penalty of life without parole and avoid the cost and complications of a capital trial altogether.

A key element in generating a system of self-reinforcement is correlation among decision-makers. There are at least four important local actors whose actions determine whether a given

homicide will lead to a death sentence: the prosecutor, the defense bar, the judge, and the jury. When such an array of actors behaves independently and their preferences are not correlated, the Central Limit Theorem shows that outcomes will be stochastic. But here the actions of one are highly dependent on the expected actions of the others. If juries will not vote for death, prosecutors will not seek it. If defense attorneys are poorly resourced and unable to stop the process, juries will be more likely to convict. If judges are enthusiastic about the death penalty, prosecutors will seek it more. In the local context, any of these factors can work in either direction: where judges raise high bars to the use of the death penalty, defense attorneys will have greater powers, juries will get more restrictive instructions, and prosecutors will know they have little chance of "getting death." Where these trends are reversed, the floodgates can open.

A second key element of our theory is that the point of reference for these local actors is their own history, not other jurisdictions in the state. In some jurisdictions, district attorneys may not have sought death in many previous cases (either because of their interpretation of whether the homicide rose to the level of the "worst of the worst" or because they did not believe they could get a local jury to vote for death, or that a judge would approve of it). Later crimes then would be subjected to a negative evaluation on the first question (is this crime worse than previous crimes where death was sought?) as well as to the second (can I succeed with the judge and jury?). In other jurisdictions, the answers to those two questions would lead to the opposite conclusion: A given crime may well be equally or more heinous than a previous one where death was sought, if death had previously been sought over 100 times (as in Los Angeles, Houston, or Dallas), and it would be clear that judges and juries do not pose an insurmountable obstacle to a death sentence. (For other studies of the consequences of self-reinforcing trends in other areas of human behavior, see for example [9–13].)

Crucially for our analytic approach, if local variability in death sentencing were driven by such stochastic factors as the homicide clearance rate, the ability to collect convincing evidence, the nature of the crimes themselves, or comparison to other jurisdictions in the state, then the hazard rate for the next death sentence in the county would be unrelated to the number of previous death sentences imposed. And, in our second test, a time-series cross-sectional analysis would find no impact for the number of previous death sentences on the likelihood of another, once that decision was properly modelled with contemporaneous predictor variables. That is, we can test directly for a self-reinforcement effect.

## Two distinct tests

In this section, we present two approaches. The first is a test for "event-dependency" in death sentencing, controlling for relevant control variables. Event-dependency models test for changes in the underlying hazard rate for the next event, controlling for risk factors related to the event as well as for the number of previous events. A common application of event-dependency models is the study of heart attacks: A patient may have a number of risk factors associated with cardio-vascular disease, but the fact that he or she has previously suffered one or more previous heart attacks increases the hazard (odds) of the next heart attack as well. Since these statistical models are well understood, we use them here to predict the hazard rate for the imposition of a death sentence. If the hazard rate increases with previous use, then, other things equal, the next event will come more quickly as the number of previous events moves from zero, to low numbers, to higher numbers.

Our second test follows a cross-sectional time-series (CSTS) approach, estimating the number of death sentences in a given county-year, across all years from 1972 through 2019, and all US counties within death-penalty states. This test uses a Zero-Inflated Negative Binomial (ZINB) model controlling for multiple possible drivers of capital punishment and, crucially, a

variable for the cumulative number of previous death sentences in that county. Like the previous estimation technique, the idea is to see if this variable exerts an independent effect on the predicted number of death sentences in a given year, in a model also controlling for other relevant factors. To run these analyses, we use Stata 13.0 and R statistical software. In all cases, our empirical results powerfully show that history matters. Supplemental Materials provide extensive robustness tests of our findings. Our key hypotheses are as follows:

1. **H1**. Controlling for relevant factors, the higher the number of previous death sentences in a county, the greater the hazard rate for the next death sentence (event-dependency).

2. **H2**: The cumulative number of previous death sentences imposed in a given county since 1972 will be a significant predictor of the number of death sentences in a given year, controlling for relevant factors.

## Event-dependency

Our first test is for event-dependency. In such a model, the odds of event $k$ are conditional on various factors as well as on the number of previous events $k-1$. Table 2 presents the relevant tests for event dependence, using the same approach as previously presented for executions in [8]. We include county-level variables as follows: population size, racial threat, homicides, and lynchings during the period from 1883 to 1930. (Research finds a geographical connection between historical lynchings and contemporary death sentences; see [14, 15].) We obtained lynching data for the Southern counties from [16] and for the remaining counties from [17]. Racial threat is defined as $100 - |70 - $percentage of population white|; background on this variable is described in [18] and [19]. (Some intuition on the variable can be gained from the following: Jefferson County, MS has a black population of approximately 86 percent; it scores among the lowest on our "threat" variable, similar to Buffalo County, SD, which is approximately 57 percent white and 0.4 percent black. On the other hand, racial threat is near its highest in counties such as Cuyahoga, OH (26 percent black); Monroe, MS (29 percent black). Racial threat and death sentences correlate at 0.16. In our Supplemental materials, we show similar results using different variables for the racial dynamics of the county in question.)

**Table 2. Conditional frailty model results for controls with death sentences as outcomes.**

|                | Coefficient | St. Error |
|----------------|-------------|-----------|
| Homicides      | .0003**     | .0001     |
| Racial Threat  | .0023       | .0017     |
| Ln Population  | .7196*      | .0181     |
| Lynchings      | .0130***    | .0034     |
| AIC            |             | 61166.3   |
| R2             |             | 0.26      |
| Max. R2        |             | 1.00      |
| Num. events    |             | 6492      |
| Num. obs.      |             | 9097      |
| Missings       |             | 26        |
| PH test        |             | .9579     |

***p < 0.001,

**p < 0.01,

*p < 0.05.

Note: State level frailty terms and event stratification included.

The results presented in Table 2 largely match our theoretical expectations. Population size, homicides, and historical lynchings increase the probability of another death sentence within a fixed time period. In addition, these models appear to fit relatively well once accounting for state-level frailty, as the maximum within-sample $R^2$ is about one. In addition, the core model of interest for sentences passes the Grambsch-Therneau test. The key quantities of interest, the baseline survivor functions, remain relatively robust and approximate the results of prior versions of the model [8].

We are not primarily interested in the control variables presented in Table 2, but rather in the question of whether, controlling for those factors, there is evidence of event-dependency. Fig 1 shows the probability of a subsequent event, *k*, given a certain number of previous events, *k-1*, based on the results presented in Table 2 (see [20, 21]).

Fig 1 gives strong evidence for event dependency. The Figure compares counties within different "strata" or groups, based on the number of previous events the county has experienced. For counties with no previous events, the bottom line shows the probability of an event over time. It increases as time goes by, of course, but the key element is that the other strata increase more quickly. If the process were not "event dependent," then the probability of the next event would be fully explained by the underlying risk-factors, or covariates, and the strata would all have the same probabilities, equal to the lowest one.

It might appear anomalous that counties in the lowest stratum would have any probability of an event, but all counties start out in the zero stratum. The statistical estimate in the model, moreover, is for a synthetic "average" county, which by definition has more homicides, population, and a different racial background than many actual counties. Therefore, the lowest stratum in the model should not be taken as an estimate for the smallest US counties; over 1,000 counties have never experienced a single death sentence over 45 years of experience due to having low population sizes and low numbers of homicides compared to the "average" county simulated in the figure. In any case, the key pattern of interest is whether the probability curves, or hazard rates, grow progressively steeper as the number of previous events increases. The Figure makes this abundantly clear: Increases are steep, indeed. Note that these are predicted values from the statistical model presented, holding all other factors constant. That is, the increasing values reflect a hypothetical situation where there is no change at all in the number of death-eligible crimes in that county. The increases are associated only with the greater history of previous use. These results strongly support H1.

## Time-series cross-sectional tests

Our second test involves applying our theory of self-reinforcement to annual county-level death sentence data. For every US county in a state with a legally valid death penalty statute, we model the number of death sentences imposed in each year from 1976 through 2018. Because death sentences are rare in most counties, our dependent variable is clustered at zero. Following similar state-level analyses (see, e.g., [14]), we employ a zero-inflated count model, which estimates two separate equations. The first equation predicts death sentence counts exceeding zero, while the second seeks to explain the odds of no death sentences in that county. The first equation is therefore a count model and the second is a logistic regression model. After statistical tests confirmed the presence of over-dispersion in the data, we determined that ZINB regression was preferable to the Poisson equivalent. We specified the models with year fixed effects and robust standard errors clustered by county.

Our measure of event dependency in the ZINB model is the county's cumulative number of previous death sentences since 1972. For any county at time *t*, this variable represents the total number of death sentences imposed in that county from 1972 to *t-1*. If there is an effect

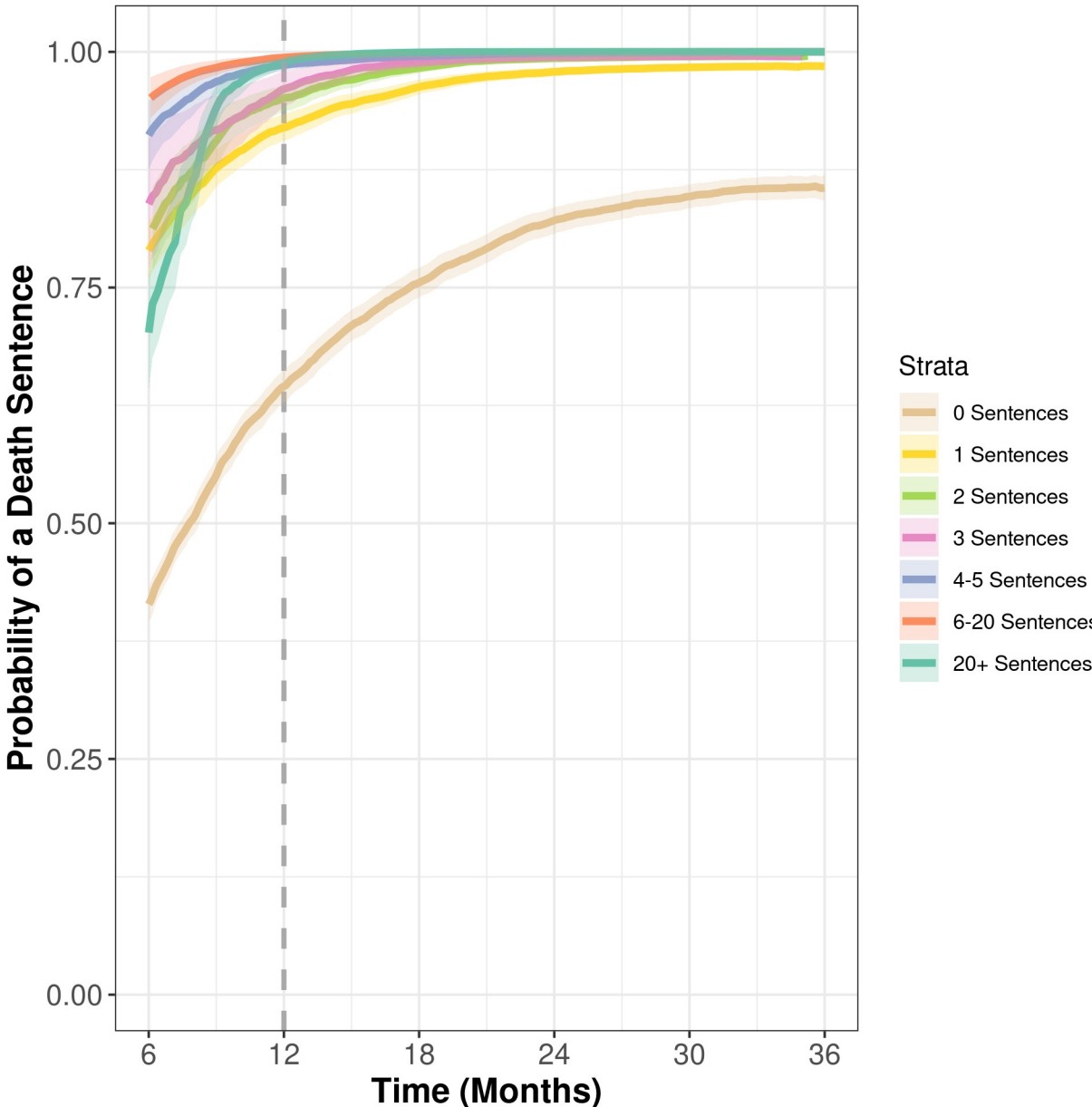

**Fig 1. Increased hazard rates for death sentences, given previous history.** Note: 1,500 bootstrap replications were used to generate survival functions and their 95% parametric confidence intervals.

associated with our theory of event dependency, then this variable will be positive and significant. We include the same county-level variables as in the previous analysis: population size, racial threat, homicides, and lynchings.

Given that counties are nested within states and criminal justice is a state function, our models also include several state-level variables. First, to account for the role of citizen ideology, we use [22]'s measure of state policy mood, where higher values indicate more liberal publics. In conservative states, not only do prosecutors face greater pressure to seek death in capital-eligible cases, but legislatures have stronger incentive to devise statutes that define a broader range of crimes as death eligible. Second, we control for the presence of a Republican

governor with a dummy variable taking a value of 1 for those years when a state has a Republican governor, and zero otherwise. Third, we control for whether a state selects its supreme court judges via partisan election [23]. Last, we denote the 11 states that constitute the South—a region that produces a disproportionate share of all death sentences [1]—with a binary indicator. (We do not include controls for any municipal ordinances, partisanship of locally elected officials, or similar factors because homicide prosecution is a state function driven by state laws and implemented by the district attorney, and state- and county-level variants of these factors, as well as time trends, are included in the models.) Table 3 shows the results.

The results for the count portion of the model indicate that population size, racial threat, and lynchings are all significant predictors of more death sentences. Note, however, that homicides, controlling for other factors, is not a significant predictor. The number of homicides is,

**Table 3. Predicting death sentences by year by county with inertia and other predictors.**

| | Death Sentence Absence | One or more Death Sentences |
|---|---|---|
| *County-level Variables* | | |
| Cumulative Death Sentences$_{t-1}$/10 | 0.03*** | 1.09*** |
| | (0.02) | (0.03) |
| Homicides$_{t-1}$/100 | 1.07 | 1.00 |
| | (0.07) | (0.02) |
| Racial Threat/100 | 0.84 | 3.02* |
| | (0.67) | (1.44) |
| Lynchings/10 | 1.12 | 1.27*** |
| | (0.22) | (0.09) |
| Ln Population | 0.72*** | 1.91*** |
| | (0.05) | (0.08) |
| *State-level Variables* | | |
| Republican Governor | 0.94 | 1.01 |
| | (0.11) | (0.06) |
| Partisan Supreme Court Elections | 0.74* | 1.00 |
| | (0.11) | (0.09) |
| Citizen Ideology/100 | 4.01 | 0.33 |
| | (4.07) | (0.21) |
| South | 0.63** | 1.00 |
| | (0.11) | (0.11) |
| Constant | 96.54*** (0.92) | 0.00*** (0.00) |
| Year FE | Yes | Yes |
| Total obs. | 102,065 | 102,065 |
| Nonzero obs. | 5,057 | 5,057 |
| Log pseudo-likelihood | -17652.32 | -17652.32 |

***p < 0.001,

**p < 0.01,

*p < 0.05

Notes: The first model predicts a value of zero death sentences, and the second model predicts the count of death sentences. In the first model, coefficients are odds-ratios and, in the count model, incidence rate ratios; these can be interpreted in the same manner. The model is a zero-inflated negative binomial regression with the county-year as the unit of analysis. Robust standard errors clustered by county are in parentheses. Counties are included only in those years where the death penalty was a legally available option in that state in that year. Each variable is rescaled by the factor indicated in order to generate coefficients that can be more easily interpreted.

however, strongly related to population size, and that variable is significant in both models. Two state-level effects are significant predictors in the model predicting no death sentences: being in the South and the presence of partisan judicial elections, both of which reduce the odds of the absence of death sentences.

In our supplemental materials we test for the impact of previous executions, rather than previous death sentences, as a driver of death sentences. Perhaps actually carrying out an execution has a more powerful impact than merely imposing a death sentence. In these alternative specifications, a previous execution does have a positive effect on the expected number of death sentences in the following year, but this is a small effect and is only marginally significant. The cumulative number of previous executions does not have a significant effect, and the lagged number of executions fades to a lack of statistical significance when we include the cumulative previous death sentences in the model. On the other hand, cumulative previous death sentences, our main measure of inertia, remains highly significant. One reason for the better statistical fit for the death sentencing variables rather than the execution-related ones is that the vast majority of death sentences are never carried out. Therefore, the link between death sentences and executions is less than one might expect.

Our substantive interest is in the variable for the cumulative number of previous death sentences, and we show a powerful impact. A value of 10 cumulative previous death sentences since 1972 is associated with a very large drop (97 percent) in the odds of no death sentences in a given year, and a nine percent increase in the odds of higher numbers. Because these effects continue for every year, their cumulative effects are much greater than the instantaneous effects shown in the table.

Table 4 provides further detail on this process, showing how the effect of inertia is particularly stark in those counties with the highest numbers of death sentences. It shows the observed

**Table 4. Observed, predicted, and simulated death sentences in top sentencing counties.**

| County | Observed | Predicted | Simulated | Percent Attributable to Previous Cases |
|---|---|---|---|---|
| Los Angeles CA | 311 | 449 | 41 | 91 |
| Harris TX | 299 | 367 | 53 | 86 |
| Philadelphia PA | 187 | 82 | 16 | 81 |
| Maricopa AZ | 179 | 121 | 27 | 78 |
| Cook IL | 157 | 199 | 40 | 80 |
| Oklahoma OK | 116 | 55 | 13 | 76 |
| Clark NV | 118 | 56 | 13 | 78 |
| Miami-Dade FL | 118 | 114 | 31 | 73 |
| Duval FL | 110 | 55 | 16 | 71 |
| Riverside CA | 110 | 45 | 13 | 71 |
| Dallas TX | 107 | 99 | 39 | 60 |
| Orange CA | 82 | 72 | 24 | 67 |
| Cuyahoga OH | 90 | 74 | 18 | 76 |
| Jefferson AL | 78 | 62 | 25 | 60 |
| Broward FL | 80 | 62 | 22 | 64 |
| Bexar TX | 77 | 64 | 29 | 55 |
| Tarrant TX | 74 | 61 | 28 | 55 |
| Shelby TN | 74 | 62 | 21 | 66 |
| Hillsborough FL | 75 | 55 | 17 | 68 |
| Hamilton OH | 70 | 48 | 11 | 76 |
| Others | 5,982 | 5,850 | 3,449 | 41 |
| Total | 8,506 | 8,053 | 3,946 | 51 |

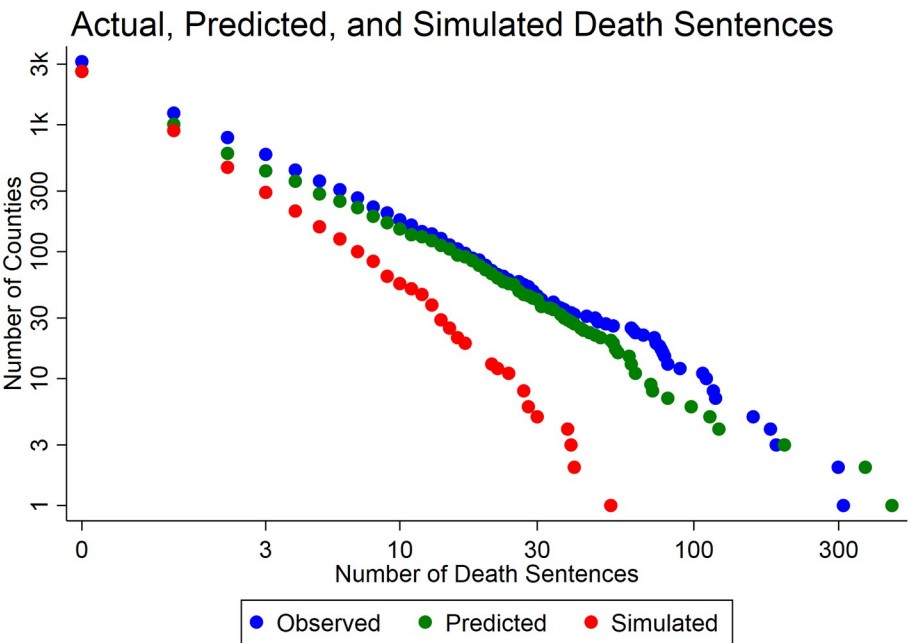

**Fig 2. Distribution of observed, predicted, and simulated death sentences by county.**

and predicted values from the model in Table 3, as well as a simulation where the values from Table 3 are used to predict the number of death sentences where the number of cumulative previous death sentences is always set to zero. The table shows the values for the top 20 death sentencing counties, as well as national totals. It shows that half of the national totals can be attributed to inertial trends and that this number is considerably higher for the most frequent users of the death penalty. This makes intuitive sense when we consider that Harris and Los Angeles counties reached 100 cumulative death sentences in 1987 and 1989, respectively. Every year since then, these two outlier counties were predicted to have substantially more death sentences than otherwise, year after year.

To illustrate the cumulative impact of this process over the entire historical period, and to show how the self-reinforcement inherent in our model tested in Table 3 generates the extreme outliers observed in Table 1, we show the full distribution of results from Table 4, including the counties not shown in the table, in Fig 2. The blue dots in Fig 2 reflect actual death sentences; the green dots reflect the predicted values from the model in Table 3; and the red dots correspond to the simulated death sentence numbers from Table 4. The comparison of the simulated to the predicted values therefore allows us to assess the impact of self-reinforcement. The comparison of the predicted to the actual values allows an assessment of the fit of the model.

Absent a system of self-reinforcement, our simulation suggest that Harris County would have 53 death sentences, not 299; Los Angeles would have 41, not 311; and no other county would have more than 40, less than one per year in the period since 1972.

## Conclusion

In 1972, US Supreme Court Justice Potter Stewart and others voting to invalidate all existing death penalty laws worried that the system was capricious and arbitrary, with those selected for the death penalty an unhappy handful randomly selected, like being struck by lightning,

out of all the eligible cases. They thought that, if any discernible pattern could be ascertained, it was racial discrimination, but they lacked clear proof of that, and they "put it to one side." Here, we have strong evidence that explains the wanton and capricious element of the death penalty: local jurisdictions separating into two camps with the vast majority never or rarely using the punishment and a small number travelling down a slippery slope of accelerating use. We also see strong evidence in favor of the racial argument: Those counties going down the path of increasing use come disproportionately from places with histories of lynching in the Jim Crow period and with higher rates of racial competition.

We can note two counter-trends the analyses we present here. Public opinion matters in two ways. First, it changes substantially over time as well as from place to place. Public opinion grew more supportive of the death penalty during the first 20 years of its renewal post-*Gregg*. Beginning in the mid-1990s, however, it has declined, and death sentences, even in those jurisdictions most likely to impose them, have declined as well [1]. Second, district attorneys are elected officials and reformist DAs have been elected in Philadelphia, PA and Harris, TX; both have pledged not to seek the death penalty in the future. So, while we have documented alarming trends with regards to capricious local habits, it does appear that even these habits can change.

## Supporting information

**S1 File. Alternative ZINB models and constructing the homicide database.**
(DOCX)

**S2 File.**
(DTA)

**S3 File.**
(RDS)

**S4 File.**
(R)

**S5 File.**
(RDS)

**S6 File.**
(DTA)

**S7 File.**
(DO)

**S8 File.**
(DTA)

**S9 File.**
(R)

**S10 File.**
(R)

## Author Contributions

**Conceptualization:** Frank R. Baumgartner.

**Data curation:** Frank R. Baumgartner, Benjamin W. Campbell, Christian Caron, Hailey Sherman.

**Investigation:** Frank R. Baumgartner.

**Methodology:** Janet M. Box-Steffensmeier, Benjamin W. Campbell, Christian Caron, Hailey Sherman.

**Project administration:** Frank R. Baumgartner.

**Supervision:** Frank R. Baumgartner.

**Validation:** Christian Caron, Hailey Sherman.

**Visualization:** Frank R. Baumgartner, Benjamin W. Campbell, Christian Caron, Hailey Sherman.

**Writing – original draft:** Frank R. Baumgartner.

**Writing – review & editing:** Frank R. Baumgartner, Christian Caron, Hailey Sherman.

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
