## [Decision Letter · Decision Letter 0]

21 Aug 2020

PONE-D-20-20778

Learning to kill: Why a small handful of counties generates the bulk of US death sentences

PLOS ONE

Dear Dr. Baumgartner,

Thank you for submitting your manuscript to PLOS ONE. After careful consideration, we feel that it has merit but does not fully meet PLOS ONE’s publication criteria as it currently stands. Therefore, we invite you to submit a revised version of the manuscript that addresses the points raised during the review process.

The reviewers and I see a great deal of merit in your work.  However, the reviewers have identified some methodological and conceptual considerations for your analysis.  First, Reviewer 1 believes that it would be helpful to show the top 25 counties in terms of death sentences per homicide (along with the county homicide rate).  Similarly, Reviewer 2 raises the question of how influential county death sentences are on the share of state executions.

Second, Reviewer 1 questions whether this story is purely about event dependence or whether your findings are a result of "differences in preferences" for the death penalty.  Is there anyway you can distinguish between these two conceptual frameworks (i.e., preference vs. dependence)?  One possible way to distinguish between these two competing frameworks would be if states or counties passed (or tried to pass) legislation on the death penalty, or if there are local policies governing the application of the death penalty.  If you find event dependence after controlling for such measures, this approach would strengthen your evidence about event dependency for Reviewer 1.

Third, at the conclusion of your article, I was left with the "so what can be done" question.  If this is a story about event dependency, what must counties do to break such dependency, or are some counties a perpetual lost cause?  Simply put, what are your policy prescriptions? 

We look forward to receiving your revised manuscript.

Kind regards,

Bryan L. Sykes, Ph.D.

Academic Editor

PLOS ONE

Journal Requirements:

Reviewers' comments:

Reviewer's Responses to Questions

**Comments to the Author**

1. Is the manuscript technically sound, and do the data support the conclusions?

Reviewer #1: Partly

Reviewer #2: Yes

2. Has the statistical analysis been performed appropriately and rigorously? 

Reviewer #1: Yes

Reviewer #2: Yes

3. Have the authors made all data underlying the findings in their manuscript fully available?

Reviewer #1: Yes

Reviewer #2: Yes

4. Is the manuscript presented in an intelligible fashion and written in standard English?

Reviewer #1: Yes

Reviewer #2: Yes

5. Review Comments to the Author

Reviewer #1: The authors first consider a repeated event model and find that higher numbers of previous death sentences reduce the expected time delay before the next death sentence. Then they consider a cross-sectional time series model for death sentences imposed in a county in a given year and conclude that the cumulative number of prior death sentences in a county is a strong predictor for the sentences imposed in a given year, and that the cumulative number of death sentences is a stronger predictor for death sentences than the number of homicides.

Data, Methodology, and Results:

The authors provide theoretical support for a model of death sentences influenced by the previous imposition of death sentences—if there is event dependency, small initial differences across counties can be transformed into a self-reinforcing system of amplified differences, as the severity of cases is usually evaluated relative previous cases. Indeed, they show that counties with similar level of homicides do have very divergent paths by presenting a table with data for the number of homicides and the number of death penalty sentences by county, and their rank in each category. They note that there are counties with similar number of total homicides but very different number of death sentences.

They use data for death sentences from 1972 to 2019, for states allowing the death penalty at the year of analysis. The US military and the federal government are excluded from the data.

They control for state in all models, as what crimes are capital eligible differs by state.

Following the same approach as Baumgarner (2018) previously followed for executions, the first test investigates event dependence by examining the hazard function across different event strata (defined as the number of death sentences). They control for the following variables: population size, a numerical measure of racial threat, homicides, and lynchings during the period from 1883 to 1930. Figure 1, showing the probability of a death sentence against time for different strata groups (0, 1, 2, 3, 4-5, 6-20, 20+ sentences), indicates very strong evidence for event dependence.

The second model tests event dependency in annual county-level death sentence data, using a zero-inflated negative binomial regression model. They model the number of death sentences imposed in each year. The same county-level variables are used as in the previous test, in addition to variables for the cumulative number of death sentences up to that year (measure of event-dependency). The following state-level variables are included: measure for state policy mood, presence of a Republican governor, control for whether the state selects its supreme court judges via partisan election, a dummy for the state being in the south. They also include year fixed effects and robust standard errors clustered by county. They find that cumulative previous death sentences has a large impact on the odds of death sentences, while the number of homicides does not.

The two tests give strong evidence for the capriciousness of the death penalty, with the counties with the increasing use of the death penalty coming disproportionately from places with troubled racial histories.

Main Criticisms:

Table 1 shows that the number of homicides and number of death sentences don’t align by presenting a table with data for homicide and death sentences cumulatively. It would be helpful to show the top 25 counties in terms of death sentences per homicide (along with the county homicide rate).

On page 8, the authors state that NYC had no death sentences during their 1995 restoration of capital punishment. I seem to recall a number of death sentences were handed down in the state and some were sought within the city. Might be useful to just note if all the actual death sentences came from the rest of the state.

On Page 13, the author states a numerical definition of racial threat, but I would have benefited from further discussion of this variable especially since it and lynchings are the two key racial variables to emerge in the Table 3 analysis. It might even be useful to identify 2 counties with different values for this measure so that one could have a better intuition of what is being captured. One might then say moving from the low level of racial threat of county X to the higher level of county Y would be expected to increase death sentences by ___.

an average level of

-The authors emphasize their finding that the number of deaths is better predicted by the county’s previous experience in imposing death penalties than by the number of homicides in the given county. They present this as evidence for capriciousness of the death penalty, saying there is little reason to believe that homicides would be more heinous if they occur in one county than another. This is true if we judge capriciousness at the state (or national) level but one might use a different model that could make this an at least somewhat rational system. If citizens vary strongly in their preferences for imposing the death penalty and elect pro-death penalty judges and prosecutors, then wouldn’t all of the patterns the authors find hold true? A liberal county would choose not to execute anyone while a neighboring conservative county might try to execute 25% of its murderers. As long as they executed the worst 25%, then they would not be acting capriciously. (Of course, they might be executing only blacks who kill whites and then this would clearly be racially problematic. Moreover, if it is the fear of black violence stemming from the legacy of slavery and lynchings that motivates the desire for the death penalty in the first instance, then the entire system would be racially inflected even if it operated racially neutrally once initiated.).

-The Governor of the state may be less important than the Mayor of the large city. I suspect that Atlanta has fewer death sentences when liberal mayors run the city regardless of who is the Governor of Georgia.

-also it might be useful to bring in the control (Google counts of searches for the n-word, of course spelled out in full) that Seth Stephens-Davidowitz used to document the racial bias by county of voters against Barack Obama. This metric has become widely adopted as one of the best indications of county variation in racial hostility and is used by Raj Chetty in his important work on how geography influences the ultimate economic success of black children.

Reviewer #2: The focus on death sentences as a county level outcome

of political (prosecutors) and legal actors is appropriate. It leaves the

issue of how much influence this has on the share of state executions from

the county would be an interesting additional issue to pursue. Also, does a previous

execution from a county issued death sentence have any measurable independent

influence on future death sentences from that county? Important because if not,

the death sentence is an independent variable and the effect of such sentences

on the rate of executions in the state is a political accident at the state level."

6. PLOS authors have the option to publish the peer review history of their article (what does this mean?). If published, this will include your full peer review and any attached files.

Reviewer #1: No

Reviewer #2: **Yes: **Frank Zimring

---

## [Author Response · Author response to Decision Letter 0]

28 Aug 2020

Thank you for the opportunity to revise our article, Learning to Kill. We were pleased that you and the reviewers found that the paper has merit, and we appreciate the questions and suggestions that you have made. Here, we enumerate the issues raised and provide our responses. We have also attached the requisite revised files, according to your correspondence of August 21.

We start with the comments you highlighted, then other comments by each reviewer.

First, Reviewer 1 believes that it would be helpful to show the top 25 counties in terms of death sentences per homicide (along with the county homicide rate).

• This was a very insightful comment; we have replaced Table 1 with a new version that also includes the death sentencing rate per 100 homicides, and the homicide rate per 100,000 population. These additions increase the power of the table, showing low correlations across the different variables. We’ve added text around the table to make this point clear. This was a very helpful suggestion and we think it enhances our points nicely. Thank you for that suggestion.

Similarly, Reviewer 2 raises the question of how influential county death sentences are on the share of state executions.

• We have tested a model now where we look at cumulative executions stemming from the county in question, as well as the lagged number of executions in the previous year, substituting (or adding) these for cumulative death sentences in our main model from Table 3. Results show that our original model is a better specification; we have added these alternative tables to the supplemental materials and written a footnote following table 3 to this effect.

Second, Reviewer 1 questions whether this story is purely about event dependence or whether your findings are a result of "differences in preferences" for the death penalty. Is there any way you can distinguish between these two conceptual frameworks (i.e., preference vs. dependence)? One possible way to distinguish between these two competing frameworks would be if states or counties passed (or tried to pass) legislation on the death penalty, or if there are local policies governing the application of the death penalty. If you find event dependence after controlling for such measures, this approach would strengthen your evidence about event dependency for Reviewer 1.

• We incorporate “citizen ideology” into our model in Table 3. This variable varies over time as well as across the 50 states and distinguishes among conservative and progressive public opinion. Further, this variable is shaped empirically like a normal distribution. Indeed, some are more conservative than others, but there are no “far outliers” that would explain such a process. Indeed, Harris County, Texas, is less supportive of the death penalty in terms of public opinion survey responses than the rest of the state (see Baumgartner et al. 2018, p. 285). We’ve added a footnote to this effect at the bottom of p. 9 in the main text.

• With regards to possibly measuring local-level variability in laws, because criminal justice is a state function, there are no relevant county-level policies. (Local ordinances may affect some things, but not homicide.) At the same time, of course, the local District Attorney plays a key role. Our modelling strategy has been to assess these locally varying characteristics (e.g., homicides, history of lynchings, racial dynamics, population size) and then see if we see a self-reinforcing process, which we do. We have added a footnote before Table 3 to make these points more clear.

Third, at the conclusion of your article, I was left with the "so what can be done" question. If this is a story about event dependency, what must counties do to break such dependency, or are some counties a perpetual lost cause? Simply put, what are your policy prescriptions? 

• We believe there are several reasonable policies, most importantly electing new DAs, and we have noted how often this has occurred, including in the highest-use counties, Philadelphia, PA and Harris, County TX, both of which elected DAs in the 2016 elections who have pledged not to seek the death penalty. Further, public opinion changes significantly over time, as is captured in our year FEs. In fact, there is a lot of variability over time. We have added relevant text in the conclusion, and we end the paper now with this very important caveat. Thank you for that reminder.

Reviewer 1 had some other points, which we have addressed as follows:

• New York City death sentences: Indeed several of the New York cases were from the city, and others from near suburbs. We’ve revised the text to delete reference to this question; there were 2 in Kings County, 1 in Queens, which are in the City, and 3 in Suffolk and 1 in Westchester, which are near suburban counties. Thanks to the reviewer for catching this error.

• More intuition on the measure of Racial threat. We’ve added relevant text in a footnote just after the first mention of this variable, before Table 2.

• The Governor of the state may be less important than the Mayor of the large city. While we can find examples either way, the death penalty is indeed a state function so we have not revised the text on this matter. We have, however, added a footnote before Table 3 to clarify.

• Google counts of searches for the n-word: We can’t do this for the full time period so have not taken this on.

In all, we thank you for the opportunity to make these improvements, and believe that these have significantly enhanced the quality of our paper. We will look forward to hearing from you.

xxx

Corresponding author

---

## [Decision Letter · Decision Letter 1]

28 Sep 2020

Learning to kill: Why a small handful of counties generates the bulk of US death sentences

PONE-D-20-20778R1

Dear Dr. Baumgartner,

We’re pleased to inform you that your manuscript has been judged scientifically suitable for publication and will be formally accepted for publication once it meets all outstanding technical requirements.

Kind regards,

Bryan L. Sykes, Ph.D.

Academic Editor

PLOS ONE

Additional Editor Comments (optional):

Reviewers' comments:

Reviewer's Responses to Questions

**Comments to the Author**

1. If the authors have adequately addressed your comments raised in a previous round of review and you feel that this manuscript is now acceptable for publication, you may indicate that here to bypass the “Comments to the Author” section, enter your conflict of interest statement in the “Confidential to Editor” section, and submit your "Accept" recommendation.

Reviewer #1: All comments have been addressed

2. Is the manuscript technically sound, and do the data support the conclusions?

Reviewer #1: Yes

3. Has the statistical analysis been performed appropriately and rigorously? 

Reviewer #1: Yes

4. Have the authors made all data underlying the findings in their manuscript fully available?

Reviewer #1: Yes

5. Is the manuscript presented in an intelligible fashion and written in standard English?

Reviewer #1: Yes

6. Review Comments to the Author

Reviewer #1: The authors responded to all of the comments appropriately, including the point about NYC death sentences, racial threat, governor versus mayor, and the Davidowitz measure of racism.

7. PLOS authors have the option to publish the peer review history of their article (what does this mean?). If published, this will include your full peer review and any attached files.

Reviewer #1: No

---

## [Editor Report · Acceptance letter]

5 Oct 2020

PONE-D-20-20778R1 

Learning to kill:Why a small handful of counties generates the bulk of US death sentences 

Dear Dr. Baumgartner:

I'm pleased to inform you that your manuscript has been deemed suitable for publication in PLOS ONE. Congratulations! Your manuscript is now with our production department. 

Kind regards, 

on behalf of

Dr. Bryan L. Sykes 

Academic Editor

PLOS ONE